# Spin-ice physics in cadmium cyanide

Chloe S. Coates[1], Mia Baise[2], Adrian Schmutzler[3], Arkadiy Simonov[1,4], Joshua W. Makepeace [1,5], Andrew G. Seel[1,6], Ronald I. Smith [7], Helen Y. Playford [7], David A. Keen [7], Renée Siegel [3], Jürgen Senker [3], Ben Slater [2✉] & Andrew L. Goodwin [1✉]

Spin-ices are frustrated magnets that support a particularly rich variety of emergent physics. Typically, it is the interplay of magnetic dipole interactions, spin anisotropy, and geometric frustration on the pyrochlore lattice that drives spin-ice formation. The relevant physics occurs at temperatures commensurate with the magnetic interaction strength, which for most systems is 1–5 K. Here, we show that non-magnetic cadmium cyanide, $Cd(CN)_2$, exhibits analogous behaviour to magnetic spin-ices, but does so on a temperature scale that is nearly two orders of magnitude greater. The electric dipole moments of cyanide ions in $Cd(CN)_2$ assume the role of magnetic pseudospins, with the difference in energy scale reflecting the increased strength of electric vs magnetic dipolar interactions. As a result, spin-ice physics influences the structural behaviour of $Cd(CN)_2$ even at room temperature.

[1] Department of Chemistry, University of Oxford, Inorganic Chemistry Laboratory, Oxford, UK. [2] Department of Chemistry, University College London, London, UK. [3] Anorganische Chemie III, University of Bayreuth, Bayreuth, Germany. [4] Department of Materials, ETH Zurich, Zurich, Switzerland. [5] School of Chemistry, University of Birmingham, Edgbaston, Birmingham, UK. [6] Department of Physics and Astronomy, University College London, London, UK. [7] ISIS Facility, Rutherford Appleton Laboratory, Harwell Campus, Didcot, Oxfordshire, UK. ✉email: b.slater@ucl.ac.uk; andrew.goodwin@chem.ox.ac.uk

The pyrochlore lattice of vertex-sharing tetrahedra is a recurring motif in many classes of geometrically frustrated materials[1–5]. Among these, systems for which each vertex is associated with an Ising variable ($e = \pm 1$, say) and which obey a constant-sum rule on each tetrahedron ($\sum_i e_i = 0$) form the particularly intriguing family of 'ices'[6]. (Cubic) water ice[7] and spin-ice $Dy_2Ti_2O_7$[8] are two examples (Fig. 1a, b); many others are known[9–12]. Common to all ice-like states is a huge configurational degeneracy—reflected in the Pauling entropy[13,14]—that in principle allows these systems to be exploited in data storage and manipulation[15]. Moreover, the constant-sum rule ($\equiv$ 'ice rule'[16]) leads to an effective gauge field that can in turn drive a variety of remarkable physics[6,17–19]. For example, violations of this rule (excitations of the gauge field) behave as emergent quasiparticles that interact with one another via an effective Coulomb potential[6,20]. These particles represent a fractionalisation of the underlying Ising variable, such that in the spin-ices they behave as magnetic monopoles (i.e., fractionalised magnetic dipoles)[21,22]. The manipulation of monopoles with external fields is thought to be a promising avenue for developing novel spintronic devices[23].

Of particular practical importance in seeking to apply this unusual physics is the energy scale that governs a given ice-like phase. How difficult is it to invert Ising states? And how strictly are ice rules obeyed? In water ice, the energies are simply too high: hydrogen-bond inversion is usually sluggish and ice rule violations are exceedingly rare ($\sim$1 ppm at 260 K)[24–26]. By contrast, spin-ices remain dynamic to very low temperatures (< 1 K), but the energy cost of defect formation is comparably small[20,27]. Hence spin-ice physics is usually constrained to the single-Kelvin regime, which is one reason why there is increasing interest in generating transition-metal analogues with stronger magnetic coupling[28].

Similarly motivated by the potential impact of identifying ice-like phases with more advantageous energetics, we study the molecular framework material cadmium cyanide, $Cd(CN)_2$, a well-known negative thermal expansion (NTE) material[29,30]. Its anticuprite structure contains cyanide ions situated on the vertices of a pair of interpenetrating pyrochlore lattices[31] (Fig. 1c). At ambient temperature, the crystal symmetry is $Pn\bar{3}m$ and the system is isostructural with high-pressure proton-disordered ice-VII[32]: the O position is occupied by Cd and the (average) H position by CN, with head-to-tail orientational disorder[33]. Solid-state NMR spectroscopy and single-crystal X-ray diffuse scattering measurements, together with density-functional theory (DFT) calculations, have collectively identified $Cd(CN)_2$ as a candidate ice[34–36]. The orientation of each individual $CN^-$ ion acts as an Ising variable, and the constant-sum rule reflects a preference for each Cd atom to bind two C and two N atoms[36], evoking the ice rules.

What is entirely unknown is whether $CN^-$ flipping is possible in $Cd(CN)_2$, and hence whether the system is capable—even in principle—of exhibiting spin-ice physics. In fact, our collective understanding of the lattice dynamics of this system is conspicuously poor. For example, on cooling to $\sim$130 K, the material exhibits a displacive phase transition that is not only uncharacterised[36], but is entirely unexpected: DFT calculations find no evidence of lattice instabilities in the parent phase[35,37]. From an experimental viewpoint, there are a number of reasons why structural and dynamical studies of $Cd(CN)_2$ are particularly complicated: one is the inability for X-ray scattering measurements to distinguish $CN^-$ orientations, especially in the presence of electron-rich $Cd^{2+}$ ions[38]; a second is the extreme sensitivity of $Cd(CN)_2$ to damage from X-ray beams, which affects the reproducibility of phase transition and thermal expansion behaviour[39,40]; a third is the (in)famously high neutron absorption cross-section of natural-abundance Cd[41], complicating both elastic and inelastic neutron-scattering measurements; and a fourth is the insensitivity of spin–lattice relaxation and the time-averaged chemical shift anisotropy for 180° jumps of the $CN^-$ ions, which renders typical NMR experiments inconclusive.

Here, we show that $Cd(CN)_2$ indeed exhibits analogous behaviour to magnetic spin-ices, and does so on a temperature scale that is nearly two orders of magnitude greater. The electric dipole moments of cyanide ions in $Cd(CN)_2$ assume the role of magnetic pseudospins, with the difference in energy scale reflecting the increased strength of electric vs magnetic dipolar interactions. As a result, spin-ice physics influences the structural behaviour of $Cd(CN)_2$ even at room temperature.

## Results

**Experimental evidence of cyanide flips.** Using a recently devised synthetic route to isotopically enriched $Cd(CN)_2$[39], we prepared a polycrystalline sample of $^{114}Cd(CN)_2$ suitable for neutron-scattering measurements. This sample has allowed us for the first time to characterise the structure of $Cd(CN)_2$ and its temperature dependence without the complications of X-ray sensitivity. Our results are shown in Fig. 2a, b. On cooling from room temperature, the $Pn\bar{3}m$ cubic unit cell of ambient-phase $Cd(CN)_2$ expands (hence NTE), until at $T_c = 130$ K a structural phase transition occurs. We find the low-temperature phase to have tetragonal $I4_1/amd$ symmetry and to be isostructural to hydrogen-ordered ice-VIII (Fig. 2c)[32]. Specifically, its crystal symmetry now allows for long-range $CN^-$ orientational order, and we do indeed find progressive ordering on cooling—evidenced by a systematic change in scattering density at the two crystallographically-distinct C/N sites—until an apparent orientational glass transition at $T_g \sim 80$ K (Fig. 2d; Supplementary Figure 1; Supplementary Discussion). No further structural transitions were observed to 10 K; note this contrasts the behaviour observed in X-ray diffraction studies, where X-ray exposure

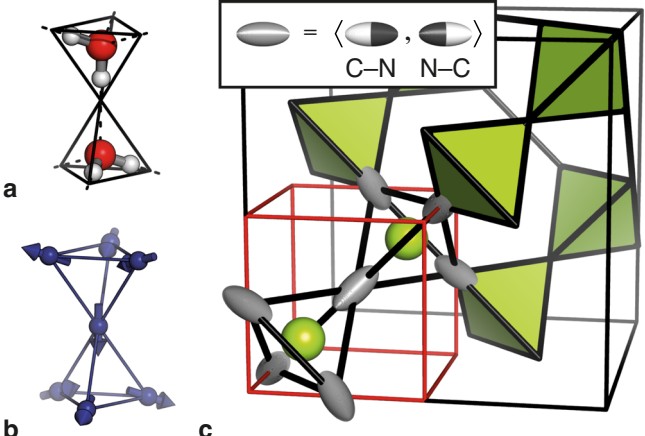

**Fig. 1 Ice rules on the pyrochlore lattice. a** Hydrogen-bond orientations in water ice and **b** magnetic moment orientations in rare-earth spin-ices both obey the same 'two-in-two-out' rule for each tetrahedral unit of the pyrochlore lattice (black lines). The same rules are thought to apply to cyanide ion orientations in $Cd(CN)_2$, the crystal structure of which is represented in **c**. Cd atoms shown as green spheres and $CN^-$ ions as ellipsoids; the unit cell (outlined in red) corresponds to one octant of the underlying pyrochlore lattice (outlined in black). For clarity, only one of the two interpenetrating pyrochlore lattices is shown. In the average structure of $Cd(CN)_2$, $CN^-$ orientations are disordered (grey). This disorder is not random: there is a preference for each Cd to bind two C atoms (white hemiellipsoids) and two N atoms (black hemiellipsoids) in an ice-like 'two-in-two-out' arrangement.

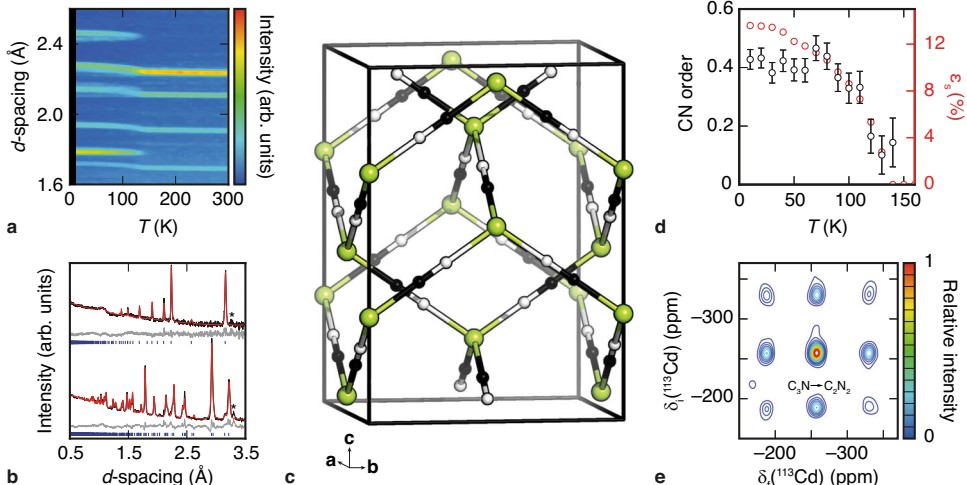

**Fig. 2 Evidence of CN⁻ flips in Cd(CN)₂. a** Intensity map of the temperature-dependent neutron powder diffraction pattern of $^{114}$Cd(CN)₂, showing the existence of a phase transition at $T_c = 130$ K. **b** Rietveld fits to the diffraction pattern at representative temperatures above (top, $Pn\bar{3}m$) and below (bottom, $I4_1/amd$) $T_c$. Data are shown in black, fits in red, difference in grey and reflection positions as blue vertical bars. The contribution from a minor Hg(CN)₂ impurity[39] is indicated by asterisks. **c** Representation of the $I4_1/amd$ crystal structure of Cd(CN)₂ at 10 K: Cd atoms are shown in green, C in white, and N in black. Thermal ellipsoids (isotropic) are shown at 50% probability. **d** Temperature evolution of the spontaneous strain (red circles) and long-range CN⁻ orientational order (black circles) determined by Rietveld refinement; error bars denote the standard errors obtained from refinement and, for the strain, are smaller than the symbols. The emergence of CN⁻ order implies that CN⁻ flips occur. The divergence of the strain and orientational order parameters marks a glass transition at $T_g \simeq 80$ K. **e** Contour plot of the $^{113}$Cd EXSY spectrum of natural-abundance Cd(CN)₂ at 60 °C showing the correlation between resonances collected before (subscript 'i') and after (subscript 'f') a mixing period of 1 s. Intensities are scaled relative to the maximum value, and contour levels are indicated on the accompanying colour spectrum. The existence of off-diagonal correlations proves the activation of CN⁻ flips, such as those which interconvert CdC₃N and CdC₂N₂ coordination environments (marked).

induces a variety of complex phases at low temperatures[40]. The non-equilibrium nature of the 80 K glass transition observed here was verified by repeated heating/cooling cycles, which showed subtle but sensible history dependencies (Supplementary Methods; Supplementary Figure 2). The crucial point of course is that we observe the emergence of CN⁻ orientational order; this is possible only if CN⁻ flips are thermally accessible, which is clearly the case for $T > T_g$. This finding is corroborated by 2D $^{113}$Cd NMR measurements taken using a natural-abundance Cd(CN)₂ sample, which show explicitly the activation of CN⁻ flips with a characteristic jump rate of about 10 Hz at 60 °C (Fig. 2e). Hence, our experimental data establish Cd(CN)₂ as a genuine candidate for spin-ice physics.

**Spin-ice model**. We proceed to determine whether a suitable spin-ice Hamiltonian can succeed in capturing the key behaviour of Cd(CN)₂, and in turn be tested against further experimental observations. Our starting point is the anisotropic Heisenberg model first proposed in ref. [42]:

$$\mathscr{H} = -J_{\text{eff}} \sum_{i,j} \mathbf{S}_i \cdot \mathbf{S}_j - \Delta \sum_i S_{i\parallel}^2, \qquad (1)$$

where the pairwise sum is over nearest-neighbour spin sites $i, j$. This model develops spin-ice behaviour at $T \sim O(J_{\text{eff}})$ for large $\Delta$ (strong single-ion anisotropy) and for ferromagnetic nearest-neighbour (effective) exchange interactions $J_{\text{eff}} > 0$. Its ground state is ordered for all finite $\Delta$, with the ordering transition temperature suppressed as the Ising limit is approached ($\Delta \to \infty$)[42]. In our mapping, the unit vectors $\mathbf{S}_i$ represent CN⁻ dipole orientations and behave as classical Heisenberg pseudospins. We use continuous rather than Ising variables because thermal fluctuations mean the Cd–CN–Cd linkages will not be entirely linear at the temperature ranges we have probed experimentally[43]. Ising-like anisotropy is introduced by the second term in Eq. (1): $S_{i\parallel}$ denotes the projection of CN⁻ orientation

vector $\mathbf{S}_i$ onto the unit vector spanning its two connected Cd centres. The parameter $\Delta$ arises from the local crystal field at the CN⁻ site and describes the barrier height to CN⁻ flipping. The exchange term in Eq. (1) will have two main contributions for Cd (CN)₂: one is a chemical bonding or covalency term—we denote this component by $J$—and the second arises from dipole–dipole interactions. Since electric and magnetic dipolar interactions have the same functional form, we can make use of the established results for the pyrochlore lattice[44,45] that long-range dipolar interactions can be effectively truncated at nearest neighbour and are described by an effective exchange term $-5D\sum_{i,j}\mathbf{S}_i \cdot \mathbf{S}_j$. Here, $D$ is the electric dipole interaction strength. Taking into account the dipolar coupling between the two interpenetrating pyrochlore lattices in Cd(CN)₂ we arrive at our model Hamiltonian:

$$\mathscr{H} = -J_{\text{eff}} \sum_{i,j} \mathbf{S}_i \cdot \mathbf{S}_j + D \sum_{i,j'} \mathbf{S}_i \cdot \mathbf{S}_{j'} - \Delta \sum_i S_{i\parallel}^2, \qquad (2)$$

where $J_{\text{eff}} = J + 5D$ (Fig. 3a). The sums in the first two terms of Eq. (2) are taken over nearest neighbours in, respectively, the same $(i,j)$ and alternate $(i,j')$ lattices. The dipolar coupling coefficients $D$ that enter these terms are identical because nearest-neighbour CN pairs lie at equivalent distances ($= a/\sqrt{2}$) whether they belong to the same or to different pyrochlore lattices (in the cubic phase); the difference in prefactors (–5 and +1, respectively) is a geometric result.

A combination of quantum chemical (QC) calculations and $^{113}$Cd NMR measurements allows us to estimate the magnitude of the various parameters in Eq. (2) relevant to Cd(CN)₂. We determined the QC energies of a range of small single- and double-network Cd(CN)₂ unit cells with different CN⁻ orientation decorations. The energies of these configurations can be interpreted in terms of $J_{\text{eff}}$ and $D$, giving $J_{\text{eff}} = 191$ K and $D = 93$ K (Fig. 3b). An alternative measure of $J_{\text{eff}}$ comes from $^{113}$Cd NMR spectroscopy, by translating the proportions of CdC$_n$N$_{4-n}$ coordination environments observed at room temperature to the

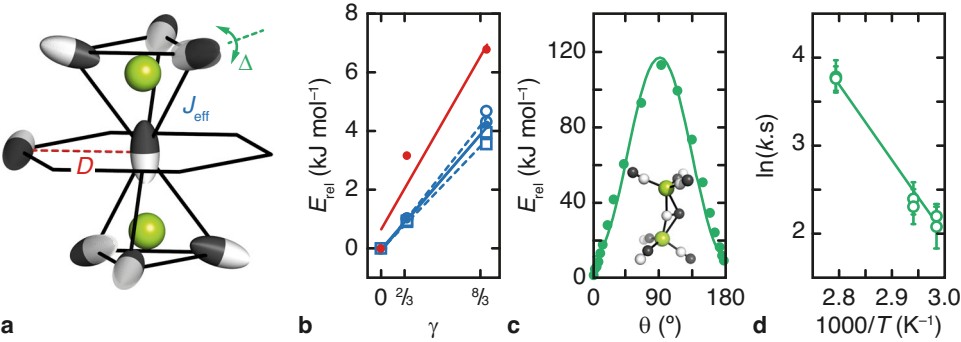

**Fig. 3 Microscopic single-ion and pairwise interaction parameters in Cd(CN)₂. a** The effective exchange interaction (strength $J_{eff}$) operates between nearest neighbours within the same pyrochlore sublattice. Dipolar interactions (strength $D$) give rise to an effective exchange interaction between nearest neighbours on alternate sublattices. The single-ion anisotropy term $\triangle$ reflects the enthalpy barrier to CN⁻ reorientations. **b** Relative DFT energies $E_{rel}$ for a series of single network (filled blue symbols) and interpenetrated (filled red symbols) Cd(CN)₂ configurations with different CdC$_n$N$_{4-n}$ coordination environments: CdC₂N₂ (geometric parameter $\gamma = \frac{2}{3}(n-2)^2 = 0$), CdC₃N/CdCN₃ ($\gamma = \frac{2}{3}$) and CdC₄/CdN₄ ($\gamma = \frac{8}{3}$). Simple geometric arguments give $E_{rel} = \gamma J_{eff}$ and $\gamma(J_{eff} + D)$ for single-network and interpenetrated Cd(CN)₂, respectively. The corresponding NMR-derived values are shown with open blue symbols (squares = ref. [34] circles = this study). **c** Nudged elastic band calculation energies for a 180° CN⁻ orientation flip (filled green symbols) and corresponding fit $E_{rel}(\theta) = \triangle\cos^2\theta = \triangle S_{\parallel}^2$ (solid green line), from which the value of $\triangle$ was obtained. The $\triangle \simeq 90°$ transition state involves a C-bridged Cd-(CN)-Cd linkage (inset) and a reduced Cd...Cd separation. **d** Arrhenius plot reflecting the temperature dependence of CN⁻ flipping rates as determined using ¹¹³Cd EXSY spectroscopy. Error bars denote the standard error in rate constants determined extracted during fitting (see Supplementary Figure 6 and Supplementary Table 7). The experimental value of $\triangle$ is given by the gradient of the linear fit (solid line).

corresponding relative free energies. The data of ref. [34] give $J_{eff} = 168$ K; our own measurements give $J_{eff} = 205$ K. The value of $D$ is harder to measure experimentally, but since the CN⁻ ion carries a dipole moment of 1.0 D, we can calculate $D = 85$ K from the crystal structure. The flipping barrier height $\triangle$ was determined computationally using nudged elastic band calculations[46] to trace the energy profile during a single CN⁻ flip. The lowest-barrier mechanism involves a C-bridged Cd–(CN)–Cd transition state and a barrier height $\triangle = 12,800$ K (Fig. 3c; Supplementary Figure 3). This value should be treated as an upper bound, given that we cannot entirely rule out alternate mechanisms involving correlated flips. 1D selective ¹¹³Cd EXSY measurements allow us to track experimentally the rate of CN⁻ flips at different temperatures; by collecting build-up curves over the range 62–85 °C we obtain $\triangle = 8800 \pm 600$ K (Fig. 3d; Supplementary Figures 4–10). The very narrow temperature window is a result of quite severe experimental constraints (see Supplementary Methods). Collectively, there is good consistency between QC and experimental results. As it happens, the key physics of Eq. (2) are surprisingly tolerant to variations in $J, D, \triangle$ values, but we take the experimental results ($J_{eff} = 205$ K, $D = 85$ K, $\triangle = 8800$ K) as representative. Remarkably, the relative energy scales of these different terms mirror those in spin-ice Dy₂Ti₂O₇, for which $J_{eff} = 3.3$ K, $D = 1.41$ K and $\triangle \sim 200$ K[44,47]; we have $D/J_{eff} = 0.41$ (0.43) and $\triangle/J_{eff} = 48$ (67) for Dy₂Ti₂O₇ (Cd(CN)₂). The key difference is that the absolute energies are ~60 times larger in Cd(CN)₂ than in an archetypal spin-ice such as Dy₂Ti₂O₇.

**Rationalisation of experimental observations.** We used these parameters to drive a series of classical Monte Carlo (MC) pseudospin simulations. Our model is subtly different to its spin-ice analogues in the sense that we have two interacting pyrochlore lattices. Nevertheless, as for the related spin-ice model[42], we also observe an ordering transition on cooling, with $T_c = 121$ K (Fig. 4a). As in ref. [42], each pyrochlore sublattice develops a nonzero magnetisation parallel to one of the cubic axes, but the two sublattice magnetisations now oppose to give a low-temperature state that is collectively antiferromagnetic. The enhancement in $T_c$ relative to the single-network spin-ice model indicates the dipolar interaction between lattices favours ordering.

On translating pseudospins into CN⁻ orientations, the corresponding (now antiferroelectric) state for Cd(CN)₂ is described by $I4_1/amd$ symmetry. So the Hamiltonian of Eq. (2) drives precisely the same phase transition we observe experimentally, in terms of both nature and temperature scale. Importantly, the same transition occurs for all sets of our QC interaction parameters; it is only the value of $T_c$ that differs (if at all). Of course, longer-range interactions, strain coupling and anharmonicity—all of which are omitted in our simple model—may mean the low-temperature $I4_1/amd$ model is not the true ground state. In fact, we may never know, since CN⁻ reorientations are experimentally inaccessible at temperatures below 80 K, and our different QC calculations also give a range of competing ice-rules-observing ground states whose energies differ by much less than this amount.

Just how well are other aspects of the structural behaviour of ambient-phase Cd(CN)₂ described by our simple spin-ice model? In Fig. 4a, we show the temperature-dependent populations of CdC$_n$N$_{4-n}$ coordination environments expected as a function of temperature from our MC simulations. We find good consistency with the trends we determine from experimental neutron pair distribution function (PDF) and magic-angle spinning (MAS) ¹¹³Cd NMR measurements, although the absolute variation over the accessible temperature range is relatively low (Fig. 4a–c; Supplementary Figure 11). In both cases, there are strong experimental constraints (e.g., long count times, high backgrounds) that limit the number of data points measurable and the temperatures to which they correspond; likewise, the low-temperature phase transition complicates interpretation of the neutron PDF for $T < T_c$. Additional support for our interpretation comes from X-ray diffuse scattering, which is indirectly sensitive to CN⁻ orientation distributions via induced Cd displacements: for example, Cd centres coordinated by two C and two N atoms displace by ~0.5 Å along a $\langle 100 \rangle$ axis towards the C atoms[36], whereas those surrounded by four C or four N remain on their high-symmetry sites. So the pseudospin orientations from our 298 K MC simulations can be used to infer a corresponding configuration of Cd displacements, which in turn allows calculation of the expected X-ray diffuse scattering pattern. We find excellent agreement with experiment, as shown in Fig. 4d.

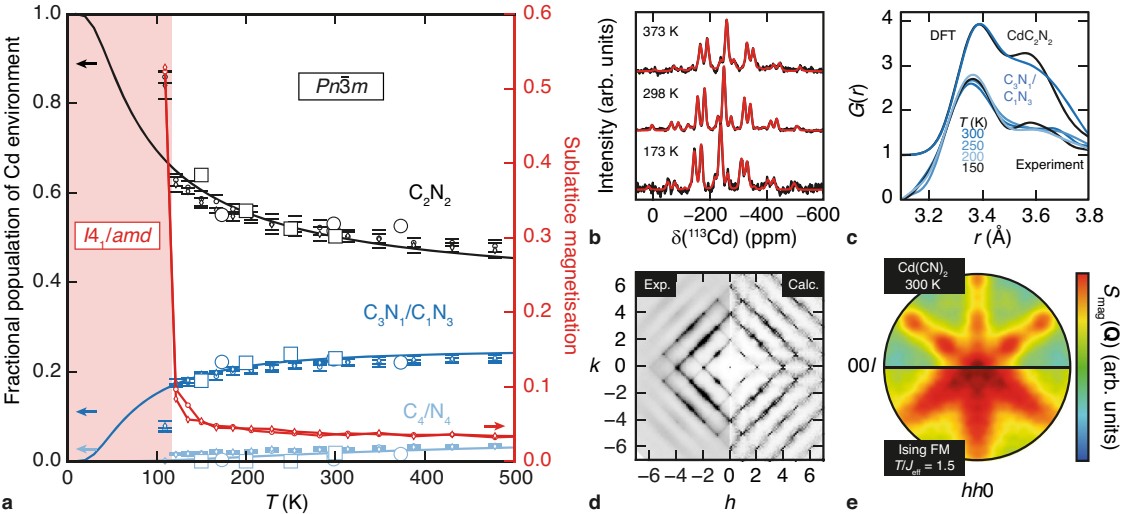

**Fig. 4 Calculated and observed structural spin-ice behaviour in Cd(CN)$_2$. a** Temperature-dependent population of CdC$_n$N$_{4-n}$ coordination environments as determined by our MC pseudospin simulations (small symbols; error bars denote the standard error measured from five independent MC simulations), neutron PDF measurements (open squares) and $^{113}$Cd MAS NMR spectroscopic measurements (open circles), compared with the non-interacting analytical result (solid lines, ref. [34]). The sublattice magnetisation (red symbols, red line)—or polarisation, in the specific context of Cd(CN)$_2$—is an order parameter for the transition between the disordered ($Pn\bar{3}m$) spin-ice state and the low-temperature ($I4_1/amd$) antiferroelectric state. **b** Variable-temperature $^{113}$Cd MAS NMR spectra (black lines) and corresponding fits (red lines) used to extract the values given in **a**. **c** Variable-temperature neutron PDF data (lower curves) and calculated PDFs from $\gamma = 0$ and $\frac{2}{3}$ QC configurations (upper curves) used to determine the PDF-derived values in **a**. **d** Single-crystal X-ray diffuse scattering pattern (($hk$0) plane) measured at 298 K and calculated from a coupled CN-orientation/Cd-displacement model based on the 298 K pseudospin configuration. **e** Effective magnetic diffuse scattering pattern (($hhl$) plane) extracted from our 300 K MC simulations by interpreting CN$^-$ orientations from a single pyrochlore sublattice as classical spin vectors with the Dy$^{3+}$ magnetic form factor (top panel). Calculated magnetic scattering for the pyrochlore Ising ferromagnet for the same relative temperature $T/J_{eff} = 1.5$ to which the Cd(CN)$_2$ data correspond (bottom panel). In both cases, fluctuations away from the spin-ice ground state broaden the pinch-point features observed in experimental magnetic diffuse scattering pattern measured, for e.g., spin-ice Dy$_2$Ti$_2$O$_7$ at 1.3 K[68]. Instead the scattering feature resemble more strongly the inelastic magnetic neutron scattering for strongly fluctuating spin-ice system such as Pr$_2$Zr$_2$O$_7$[50].

The corresponding neutron-scattering calculation, for which there as yet no comparable single-crystal experimental measurements, is provided as Supplementary Figure 12.

## Discussion

So, the key ingredients for magnetic spin-ices—namely, single-ion anisotropy and magnetic dipolar interactions—are mapped onto their electrostatic analogues in Cd(CN)$_2$ with a concomitant transformation in energy scale. The phase behaviour of this non-magnetic system—a 'dipolar structural spin-ice'—obeys a closely related physics to that of the spin-ices themselves. Ordinarily, key experimental signatures of the spin-ice state are measurements of the residual Pauling entropy and/or observation of a set of 'pinch-points' in the magnetic diffuse scattering pattern that arises from the underlying gauge symmetry[48,49]. The stability window of cubic Cd(CN)$_2$ is unfortunately too high in energy ($T_c \simeq J_{eff}/2$) for the system ever to delve deep within the spin-ice state, and so there is no scope for observing either feature here. In any case, we do not yet have access to single-crystal $^{114}$Cd(CN)$_2$ samples that would allow the issue of pinch-point scattering to be tested experimentally. What we can do is to calculate from our MC simulations the effective magnetic (neutron) diffuse scattering for an analogous spin model; the corresponding room temperature pattern is shown in Fig. 4e. The purpose of this calculation is to compare the pairwise psuedospin correlation functions that develop in our model, as reflected in the effective magnetic scattering function $S_{mag}(\mathbf{Q})$, with those of conventional spin-ices. As expected, there are no pinch-point features, but the scattering is far from featureless and its form reflects the presence of spin-ice correlations even at room temperature. In fact, the scattering

shows even more structure than that for the simple Ising pyrochlore ferromagnet (spin-ice ground state) at the same relative temperature $T \simeq 1.5 J_{eff}$, and resembles the inelastic contribution for strongly fluctuating spin-ices such as Pr$_2$Zr$_2$O$_7$[50].

Other aspects of spin-ice physics are also relevant to Cd(CN)$_2$. The C-rich or N-rich Cd coordination environments assume the role of emergent monopoles[49]; they represent a fractionalisation of the molecular cyanide ion and must interact via a Coulomb potential[6] (perhaps unsurprising given the electrostatic origin of the spin-ice state in Cd(CN)$_2$). One might hope to manipulate monopole distributions by the application of an external electric field. Remarkably, the soft phonon mode responsible for the $Pn\bar{3}m$ to $I4_1/amd$ transition is also a manifestation of spin-ice physics[42]. This explains why ordered models do not show any lattice instabilities in DFT calculations, and points to the intriguing interplay between (long timescale) CN$^-$ flips and (short timescale) lattice dynamics that is itself reminiscent of extreme rotovibrational coupling[51]. Inelastic neutron-scattering measurements of the dynamics of Cd(CN)$_2$ are an obvious avenue for future study.

We finish by asking: to what extent might this behaviour be expected to generalise to other, related materials? Zn(CN)$_2$, for example, is isostructural to Cd(CN)$_2$ but does not show the same phase complexity[29]. Our QC calculations explain why: $J_{eff}$ is smaller and $\triangle$ larger than for Cd(CN)$_2$—in each case, a consequence of increased covalency and higher charge density at the Zn site. So, ice rules are not so strongly enforced, nor are CN$^-$ reorientations thermally accessible; indeed the increase in $\triangle$ also reflects the reduced NTE effect in Zn(CN)$_2$ observed experimentally[29]. Single-network $s$-Cd(CN)$_2$[52] is likely to be the more interesting analogue since its behaviour may be describable

in terms of the simpler and more strongly frustrated $Dy_2Ti_2O_7$ Hamiltonian of Eq. (1). If the parameters $J_{eff}$ and $\triangle$ are comparable to those in $Cd(CN)_2$ itself, one expects the system never to order experimentally, since the phase transition temperature should be suppressed below the onset of orientational glass formation. This is consistent with the absence of any phase transition in variable-temperature (100–300 K) X-ray diffraction measurements[52]. $Cd(CN)_2$ also forms a very large array of host–guest structures, many of which are based on the pyrochlore lattice[53]. This presents the unexpected possibility of tuning spin-ice behaviour via guest (de)sorption. The substitution of $CN^-$ for $Br^-$—as explored historically in the context of alkali cyanide quadrupolar spin-glass analogues[54]—will have an effect equivalent to doping a spin-ice with non-magnetic impurities. Likewise, pressure is an as-yet unexplored variable for spin-ice physics that is now suddenly accessible given the shift in temperature scale. One way or the other, our study has reinforced the concept that materials with strongly-correlated structural disorder can mirror the remarkable physics of exotic electronic phases[5]. But it demonstrates also how the theory that underpins our understanding of the latter helps rationalise the phase behaviour of the former. Noting the empirical mapping between symmetry breaking in $Cd(CN)_2$ and the VII/VIII proton-ordering transition in water ice[55], for example, one might reasonably ask whether the phenomenology of spin-ices may yet shed light on the physics of their fundamentally important parent: water ice itself.

## Methods

**Synthesis**. We prepared a polycrystalline sample of isotopically enriched $^{114}Cd$ $(CN)_2$ following the method described in ref. [39]. In all, 1 g of $Hg(CN)_2$ and a stoichiometric excess of $^{114}Cd$ metal (1 g) were added to one arm of a custom-made glass N-cell. Anhydrous $NH_3$ gas (30 mL liquid volume) was condensed onto the mixture and stirred for 6 hours in an acetone/dry-ice bath with the temperature maintained between 240 and 250 K. The mixture was filtered through the porous frit separating the two Schlenk tubes of the N-cell under flowing ammonia gas to remove insoluble Hg. The resulting solution was allowed to evaporate, yielding a polycrystalline sample of $^{114}Cd(NH_3)_2[^{114}Cd(CN)_4]$. This solid was then heated at 80°C for 24 h to yield $^{114}Cd(CN)_2$ as a white powder. The synthesis was carried out in three batches. The sample contains <2 wt% contamination with $Hg(CN)_2$.

The $Cd(CN)_2$ sample used for NMR measurements was prepared using natural-abundance Cd and the conventional synthesis route as described in ref. [29]: stoichiometric quantities of $Cd(NO_3)_2$ and $K_2Cd(CN)_4$, each prepared as aqueous solutions, were combined and the resulting solution allowed to evaporate slowly. Single-crystals of $Cd(CN)_2$ appear as the first precipitate; these were harvested, washed with $H_2O$ and ground to a fine powder.

**Neutron diffraction and total scattering**. Time-of-flight powder neutron diffraction data were collected using the POLARIS diffractometer at the ISIS pulsed neutron and muon source, Rutherford Appleton Laboratory, UK[56]. In all, 1.0504 g of a polycrystalline sample of $^{114}Cd(CN)_2$ was loaded into a 6 mm diameter thin-walled cylindrical vanadium sample can to a depth of 4 cm, which was then placed into a AS Scientific helium flow cryostat at a temperature of 300 K. A RhFe sensor was attached to the outside of the vanadium sample can to monitor the sample temperature throughout the experiment. Data collection consisted of a series of short duration (∼10 min each) diffraction patterns every 10 K as the sample was cooled (cooling rate ∼0.5 K min$^{-1}$) for Rietveld analysis (average crystal structure and unit cell determination), interspersed with much longer duration (∼ 6 h each) data sets at 300, 250, 200, 150, 100, 50 and 10 K for total scattering analysis. Data reduction and normalisation were carried out using the MantidPlot software[57] with the final processed files from the five Polaris detector banks covering a scattering range $0.5 < Q < 50$ Å$^{-1}$. Total scattering data were corrected and the pair distribution function $G(r)$ was generated using the GUDRUN software[58].

**Single-crystal X-ray diffuse scattering**. Single-crystal X-ray diffuse scattering data were measured on the beamline BM01 at the European Synchrotron Radiation Facility (ESRF), Grenoble, France. The measurement was performed at the wavelength 0.6975 Å using a Pilatus 2 M detector. The experiment consisted of two 360° scans: one optimised for the Bragg peaks and the second with stronger primary beam flux optimised for the diffuse scattering. The crystal orientation was determined using the programme XDS[71] and the diffuse scattering reconstruction was performed in the programme Meerkat (available at https://github.com/aglie/meerkat). Bragg peaks were removed using the punch-and-fill method[59].

**Solid-state magic-angle NMR**. $^{113}Cd$ quantitative and exchange NMR spectra were acquired on an Avance III HD spectrometer (Bruker) at an external $B_0$ field of 9.4 T ($v_0 = 88.7$ MHz) using commercial 3.2 mm and 1.9 mm MAS triple resonance probes (Bruker). Further 1D exchange experiments were measured with a commercial 4 mm MAS triple resonance probe (Bruker) on an Avance II spectrometer (Bruker) at an external $B_0$ field of 7.1 T ($v_0 = 66.58$ MHz). The spinning speed was set to values between 8.25 and 22.222 kHz, so ensuring that no overlapping of signals or sidebands was observed.

Quantitative MAS spectra were obtained using single-pulse excitation with a 90° pulse of 2.5 μs. The recycle delay was adjusted so that the recovery of the longitudinal magnetisation was larger than 90%. Exchange measurements were acquired by a sequence of three 90° pulses (Supplementary Figure 4a) with a length between 1.75 and 2.5 μs. For the 1D exchange spectra, the first pulse was replaced by a Q5 Gaussian pulse cascade[60] (Supplementary Figure 4b) of 2 ms (9.4 T) and 2.7 ms (7.1 T) duration to selectively excite one of the resonances by setting the transmitter frequency offset accordingly. The $t_1$ delay was set to very short values of up to 2 μs. Exchange spectra were recorded with a 16-fold phase cycling, recycle delays were adjusted to provide at least 80% of the longitudinal magnetisation in equilibrium.

The $^{113}Cd$ chemical shifts are reported with respect to $(CH_3)_2Cd$, using Cd $(ClO_4)_2$ as a secondary reference. The variable-temperature measurements were carried out using dry nitrogen for both drive and bearing; a constant stream of cold nitrogen aimed at the centre of the rotor allowed adjustment to the desired temperatures. For the exchange experiments, dry air was used instead of nitrogen. To reduce temperature gradients, the sample inside the 3.2 mm rotor was sandwiched between two layers of sodium chloride. For the 4 mm probe, a CRAMPS rotor was used. The resonances in NMR spectra were fitted by a Pseudo-Voigt line shape.

**QC calculations**. Structure optimisation was performed primarily in the Quickstep module in CP2K (available at: https://www.cp2k.org/)[61], with atoms and cell parameters allowed to relax simultaneously, and a double-ζ basis set[62] and PBE-D3 functional with an 850 Ry cutoff (D3 here refers to the scheme of ref. [63,64]. Self-consistent field cycles were converged to $10^{-6}$ eV and forces on atoms to 0.01 eV Å$^{-1}$ or less. A 4×4×4 supercell was used for the optimisation within a gamma point sampling regime.

Influenced by the approach of ref. [35], we calculated energies for six separate configurations, constructed as follows. The first corresponded to the highest-symmetry version of the $Cd(CN)_2$ structure with all Cd centres in $CdC_2N_2$ coordination environment; the second to that with $CdCN_3/CdC_3N$ environments; and the third to that with $CdC_4/CdN_4$ environments. Then, for each of these three cases we generated an additional configuration containing a single $Cd(CN)_2$ network (rather than the two interpenetrating nets in the native structure). These various configurations can be associated with the geometric parameter $\gamma = \frac{2}{3}(n-2)^2$, where $n$ is the number of C atoms in the Cd coordination sphere. It is straightforward to show that the configurational energy per mol Cd relative to that of the $CdC_2N_2$ ground state should vary as $\gamma J_{eff}$ for the single-network configurations, and as $\gamma (J_{eff} + D)$ for the interpenetrated networks. Consequently, it is possible to extract from the QC energies estimated values of $J_{eff}$ and $D$. The corresponding energies and derived $J_{eff}$, $D$ parameters are listed in Supplementary Table 1.

The low energy pathway for cyanide reorientation was obtained using CP2K[61] and with the climbing-image nudged elastic band (CI-NEB) method. It was found that nine replicas along the pathway were sufficient to capture the barrier height (Supplementary Figure 3 and Supplementary Table 1); additional calculations using 13 and 19 images gave the same barrier as reported with a difference of <0.05 kJ mol$^{-1}$.

**MC simulations**. The MC simulations described in the text were carried out as follows, making use of custom code based on that described in ref. [65]. Each MC configuration comprised 2048 pseudospins, arranged on the $CN^-$ ion positions of an 8 × 8 × 8 supercell of the ambient-phase $Pn\bar{3}m$ $Cd(CN)_2$ unit cell. These pseudospins were unit vectors, and represented the orientation of a corresponding $CN^-$ ion. MC simulations were initialised by assigning random orientations to each pseudospin and the configurational energy calculated according to Eq. (2). Each MC step involved determining a candidate pseudospin reorientation, which was then accepted or rejected according to the Metropolis MC criterion. The simulation temperature was initialised at 1000 K, and an equilibration time determined according to loss of autocorrelation in the pseudospin correlation function (we equilibrated configurations for 10 times as many steps as that estimated for decorrelation). Values at each temperature were averaged over five successive decorrelated MC configuration states before the temperature was reduced by a factor of 0.1. The system was again allowed to come to equilibrium, and the process repeated at the reduced temperature. As in previous MC studies of spin-ices[42], we found the simulation struggled to equilibrate after cooling through its ordering phase transition. Of course, this did not prohibit us from identifying the existence and nature of the phase transition. Our MC simulations were repeated in full for five independent runs, and final values were taken as an average over these different runs.

Although our key results are based on the experimental values of $J_{eff}$, $D$, $\triangle$, we found that all combinations of these parameters obtained in our additional QC calculations (Supplementary Discussion, Supplementary Table 1) gave qualitatively similar results, differing only in the numerical value of $T_c$. We did not, however, observe a transition for the $Zn(CN)_2$ parameter set (Supplementary Table 2), which is consistent with the absence of any experimentally-observed phase transition in this material[29].

**Diffuse scattering calculations**. Geometry relaxation of $8 \times 8 \times 8$ supercells of Cd$(CN)_2$ was performed using the General Utility Lattice Programme (GULP)[66], in order to simulate diffuse scattering based on a coupled $CN^-$-orientation/Cd-displacement model. Here, the basic idea was to use the $CN^-$ orientations determined from our MC simulations to infer a corresponding set of Cd displacements as outlined in ref. [36]: a Cd atom in a $T_d$-symmetric $CdC_4$ or $CdN_4$ environment is not expected to displace strongly, but one in the $C_{2v}$ $CdC_2N_2$ environment displaces by ~0.5 Å towards one edge of the coordination environment (for completeness we note that we saw the same effect in our QC calculations). As X-ray scattering measurements are insensitive to $CN^-$ orientations, but sensitive to Cd displacements, we are then able to check whether the observed X-ray diffuse scattering patterns[36] are nonetheless indirectly rationalisable in terms of the $CN^-$ interaction model developed in our study.

The geometry optimisation was performed using GULP operating at constant pressure. The force-field parameters used were: harmonic bond potentials for Cd–(C/N) ($k = 20$ eV Å$^{-2}$, $r_{eq} = 2.155$ Å) and C–N ($k = 60$ eV Å$^{-2}$, $r_{eq} = 1.148$ Å); charges of $+2$, $-1$ and $0$ $q_e$ for Cd, C and N, respectively; and a linear three-body term for the Cd–(C/N)–(N/C) bonds ($k_{three} = 3$ eV, $i_{sign} = 1$, $n = 1$). The corresponding X-ray diffuse scattering pattern was generated and the ($hk0$) slice is as shown in Fig. 4d. Note that the form of the diffuse scattering is insensitive to the specific force constants used; the values we used here were chosen to give sensible relaxed structures and magnitudes of Cd off-centreing. For completeness we also calculated the corresponding single-crystal neutron diffuse scattering pattern (Supplementary Figure 12), which—in the absence of large single-crystal samples of isotope-enriched Cd$(CN)_2$—is yet to be measured experimentally.

**Effective magnetic neutron diffuse scattering calculations**. The effective single-crystal magnetic diffuse scattering pattern shown in Fig. 4e was calculated using the SPINDIFF software, which is part of the SPINVERT distribution[67]. The input configurations were those generated in the MC simulations described above, carried out at 300 K. Pseudospins were assigned the magnetic form factor of $Dy^{3+}$. Because the Cd$(CN)_2$ structure contains two interpenetrating pyrochlore lattices, we calculated the effective scattering for each lattice independently and averaged over the two sets of calculated scattering intensities. This allowed direct comparison with the experimental magnetic diffuse scattering data for $Dy_2Ti_2O_7$ as shown.

## Data availability

The neutron-scattering data are available from the ISIS facility with reference https://doi.org/10.5286/ISIS.E.RB1720378. All other data sets generated during and/or analysed during the current study are available from the corresponding authors on reasonable request.

## Code availability

All custom code used in this study was developed using widely available algorithms. Copies of the actual code used can be obtained upon request.

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

## Acknowledgements

The authors gratefully acknowledge funding from the E.P.S.R.C. (Grants EP/G004528/2, EP/L000202, EP/R029431), the E.R.C. (Grant 788144), the German Science Foundation (Grant SFB840 C1), the Leverhulme Trust (Grant RPG-2015-292), the Swiss National Science Foundation (Fellowships to A.S., nos. P2EZP2_155608 and PZ00P2_180035) and St John's College, Oxford (Fellowship to J.W.M.). Calculations were performed on ARCHER, the UK National Supercomputing Service (http://www.archer.ac.uk) using time allocated by the Materials Chemistry Consortium. Single-crystal X-ray diffuse scattering measurements were performed on beamline BM01 at the ESRF, Grenoble, France. We are grateful to Dmitry Chernyshov (ESRF) and Hanna Boström (Stuttgart) for their assistance in using the beamline. We gratefully acknowledge useful discussions with Lucy Clark (Birmingham) and Joseph Paddison (Oak Ridge National Laboratory).

## Author contributions

C.S.C., M.B., B.S. and A.L.G. designed the research. C.S.C., J.W.M. and A.G.S. synthesised the materials. C.S.C., M.B., R.I.S., H.Y.P., D.A.K. and A.L.G. measured and interpreted the neutron-scattering data. A.Sc., R.S. and J.S. measured and interpreted the NMR data. M.B. and B.S. performed and interpreted the QC calculations. C.S.C. and A. Si. Measured and interpreted the X-ray scattering data. C.S.C., A.Si. and A.L.G. developed and implemented the MC model. C.S.C., M.B., A.Si., B.S. and A.L.G. wrote the manuscript, with input from all authors.

## Competing interests

The authors declare no competing interests.
