## [Peer Review File · Nature Communications]

REVIEWER COMMENTS

Reviewer #1 (Remarks to the Author):

In this manuscript a system of frustrated electric dipoles (“dimers”) in $\text{Cd}(\text{CN})_2$ -- argued to be analogous to the physics of magnetic dipoles in spin ice materials -- is presented. Due to the larger energy scale of the electric dipole interaction (vs. the magnetic dipole interaction) elements of this physics appear even at room temperature, much higher than the $O(1\text{K})$ scales typical for the rare-earth based spin ice systems. The presence of this “spin ice” physics is primarily supported by a combination of neutron PDF and Cd NMR measurements, which indicate that the population of local CN dimer configurations tracks well with expectations based on an spin ice (Ising) model down to $\sim 130\text{K}$ or so. The dimer-dimer interactions being “ice-like” is supported by quantum chemical calculations. Below $\sim 130\text{K}$ a structural transition into an anti-ferroelectric state prevents exploration of any low temperature aspects of this ice physics. Some elaboration on the spin ice analogy is made through comparison of single crystal X-ray diffraction to theoretical expectations, as well as modelling what neutron scattering would reveal in a magnetic system analogous to $\text{Cd}(\text{CN})_2$.

Overall my impression of this work is quite positive. This material $\text{Cd}(\text{CN})_2$ presents an important opportunity to explore frustration (and spin ice physics in particular) at much higher temperature regimes and outside the usual set of magnetic materials. It will thus likely be of interest to those studying spin ice physics and frustrated magnets more broadly, as well as those interested in these kinds of structurally disordered/correlated materials.

The key result seems to be the characterization of this material above its structural transition. The manuscript makes a fairly convincing case that the interactions between the dimers are ice-like and that these dimers are at least somewhat dynamic above $\sim 100\text{K}$ or so. The theoretical analysis is consistent with this, though it is not so clear how reliable these kinds of quantum chemical calculations are in estimating these parameters. The range of methods presented in the SI make it clear that different ab-initio methods agree qualitatively, but not quantitatively, especially for the dipolar energy scale.

Unfortunately, the (structural) phase transition at $\sim 130\text{K}$ cuts off access to the low temperature physics, with the paraelectric phase disappearing at temperature scales comparable to the effective Ising exchange in the spin ice model, due to the presence of the two pyrochlore sublattices which are coupled through (at least) dipolar interactions $\sim O(100\text{K})$. This is a major difference from spin ice materials, where the limiting factor is the intra-sublattice dipolar interaction which, on its own in $\text{Cd}(\text{CN})_2$, would only result in ordering at much lower temperatures $\sim O(10\text{K})$. In other words this model, with its two sublattices, is much less frustrated than the usual dipolar spin ice.

While this is a real deficiency, I do not think it is a fatal one, as there are many potential avenues in perturbing or modifying this material that might alleviate this issue (e.g., as discussed by the authors, applying pressure, elemental substitution, etc). In addition, this two sublattice model has not been studied in much detail, and so may hold some surprises.

The authors further raise a number of interesting questions regarding the dynamics of the CN dimers and the relation of the conventional phonon modes to the unconventional excitations of the "ice" degrees of freedom (in this case the electric monopoles). These are likely to provide good starting points for future experimental and theoretical studies of these types of systems. I think the comments about single-network Cd(CN)₂ are particularly promising, since the issue with ordering might be avoided (or at least strongly suppressed). The relation of this physics to its observed NTE is also intriguing.

In terms of the novel results demonstrated in the work, its appeal to a broad range of researchers and its potential to motivate further work, I believe this satisfies the criteria for publication in Nature Communications and thus recommend acceptance.

Reviewer #2 (Remarks to the Author):

The authors consider the phase behaviour and order associated with CN- orientations as well as Cd displacements to argue that Cd(Cn)₂ likely displays phenomena akin to spin-Ice physics, but at relatively high temperature compared with that displayed by well-studied magnetic insulators with spin-Ice ground states.

This is a very interesting analogy to consider and expand upon, with the orientation of the CN- molecules playing the role of the large, Ising-like magnetic dipole moments in magnetic insulators such as Ho₂Ti₂O₇ and Dy₂Ti₂O₇. This geometrically frustrated ground state has generated much excitement, both in the classical spin Ice problem and its quantum analogue, quantum spin ice. Of course, it is not so surprising that a structural analogue to such a magnetic problem could exist, as our understanding of these magnetic insulators is based on analogy with the physics of "real" water ice.

As the authors discuss, the smoking gun signatures for spin ice correlations and spin ice physics in magnetic insulators have been the observation of pinch-point diffuse scattering in single crystal scattering experiments and a residual Pauling entropy associated with the heat capacity in these systems. I think it is fair to say that were convincing data of this form available and presented, we would likely be discussing a contribution to Nature.

As it is, such smoking gun signatures are not available, at least in part because single crystals appropriate for neutron scattering studies are not available, and the temperature scale of the phenomenon in $\text{Cd}(\text{CN})_2$ is as high as it is. Nonetheless, this is a very well written manuscript that sets out the case that indeed CN- flips do occur, and that some orientational order of the CN- molecules develop in the temperature range of interest. They then map the available data onto a minimal effective spin Hamiltonian and show that the corresponding state describing $\text{Cd}(\text{CN})_2$ should be related to classical spin ice over a reasonable high temperature regime. While I have a couple of questions, I believe that this is an interesting account of a new and topical problem, and that it is appropriate to Nature Communications, after the relatively minor points have been addressed.

My relatively minor points are:

1- The authors show x-ray diffuse scattering attributed to displacements of the Cd and dependent on the details of the local CN- orientations around the Cd. Presumably the size of the Cd displacements was determined (perhaps this is in the supplemental info - if so I would highlight this in the main manuscript), and this should be explicitly mentioned. I also assume that there are no corresponding displacements associated with the magnetic spin ice problem. Would such displacements, and/or local distortions of the tetrahedra influence the predicted diffuse neutron scattering and pinch points as seen in Fig. 4 e)?

2- There has been significant recent interest in insulating pyrochlore magnets based on transition metal magnetism, rather than rare earth magnetism, and much of this has been motivated by the same motivation presented here - to access spin ice like physics at higher than Kelvin and sub-Kelvin temperatures. The system NaCaX_2F_7 with $\text{X}=\text{Co}^{2+}$ and Ni^{2+} have been recently studied, with some evidence for pinch point correlations in $\text{NaCaNi}_2\text{F}_7$. Of course, these systems have a disordered "A" pyrochlore sub lattice, nonetheless, their relevant temperature scales are much elevated compared with, say, $\text{Dy}_2\text{Ti}_2\text{O}_7$, and it may be worth mentioning these. The relevant publications I know of are:

K.A. Ross et al, PRB, 93, 104433, 2016; Plumb et al, Nature Physics, 15, 54, 2019; Zhang et al, PRL, 122, 167203, 2019.

Reviewer #3 (Remarks to the Author):

Spin-ice physics in cadmium cyanide.

This article describes x-ray and neutrons diffraction, MAS NMR measurements and Monte Carlo calculations on the NTE material Cd(CN)₂. The motivation according to the authors is to find ice-like phases at more accessible temperatures. The manuscript is well organised and written.

However i cannot recommend publication in Nature Comm.

The basic premise of the article is misleading, that spin-ice physics (such as that seen in Dy₂Ti₂O₇ and H₀₂Ti₂O₇) occurs in this system. The authors do show very convincingly that proton ordering ice-rules and charge-ice dynamics are observed. However the experimental evidence used to map this onto the spin-ice behaviour is not sufficient to support the claim of "spin-ice" physics.

Not to mention that there is simply no spin present in Cd(CN)₂. Granted we often use classical spin models, but nevertheless magnetic spin dipole moment is not equivalent to an electric dipole. Spin is coupled to angular momentum, defined by spin operators and quantum numbers m_i , obeys selection rules, parity, etc.

Why force a round peg into a square hole ?

here are a few major complaints:

1) After an excellent introduction, the authors then try to describe Cd(CN)₂ using the Heisenberg Hamiltonian of ref 44. But immediately this does not work, because Cd(CN)₂ consist of two identical interpenetrating pyrochlore lattices, with each ion thus feeling twice as many nearest neighbours, and dipole forces along different directions.

2) In one of the most important papers (ref 14), the canonical spin-ice system Dy₂Ti₂O₇ was found to be an analog to cubic ice when heat capacity measurements showed an excess entropy that could be explained by the ice-rules first described by Pauling.

The authors explain that this cannot be measured because there is a structural phase transition at 130K, and thus cannot enter deep into the “spin-ice” phase, presumably to integrate the heat capacity. This is a pity. But even if it were possible, this would only show that Cd(CN)₂ obeys the ice-rules.

3) For the same reasons as above, and the fact that large single crystals are not available, magnetic diffuse scattering experiments could not be made in order to observe the “pinch-points” as seen Dy₂Ti₂O₇ and H₀₂Ti₂O₇. But even if there was a large single crystal, and for some reason there was no phase transition, what would magnetic diffuse scattering reveal on a sample that is simply not magnetic ? (not to mention the problems to measure it with Cd neutron absorption!)

Nevertheless the authors offer as “proof” Monte Carlo simulations of what magnetic diffuse scattering might look like compared with other systems in figure 4e. In particular they compare the simulation with “magnetic diffuse scattering” of Dy₂Ti₂O₇ and Pr₂Zr₂O₇. But the scattering pattern shown for Dy₂Ti₂O₇ is not magnetic diffuse scattering, it is just diffuse scattering, and for Pr₂Zr₂O₇ the measurements were at fixed energy 0.25meV, so it is not the same thing. Finally for Monte Carlo results they adopt the magnetic form factor of Dy. This makes no sense.

4) In this paper, the dynamics have been probed by exchange NMR spectroscopy (EXSY) described in some detail in the SI. From the analysis, thermal activation of the CN dipoles is deduced. However the results are based on only three temperatures 62, 67 and 85C ! Why ? In the main text it is mentioned “quite severe experimental constraints (see SI)” but this is not explained. Is the relaxation too difficult to measure? is it long below 60 C? too fast above 80 C ?

This is the main experimental conclusion of the paper, that thermal activation in Cd(CN)₂ is similar to spin-ice.

But this is by no means conclusive.

The dynamics in spin-ice is fascinating and has been instrumental in understanding the underlying physics. The dynamics of canonical spin-ice materials have been probed by ac-susceptibility and relaxation measurements over many decades in frequency and from room temperature down milli kelvin temperatures. There is indeed thermal activation at high temperature, then as temperature is lowered a plateau develops where tunnelling is presumably occurring, then at lowest temperatures the dynamics are dominated by emergent magnetic monopoles.

There is no analog with Cd(CN)₂

5) Finally the authors mention that “the C-rich or N-rich Cd coordination environments assume the role of emergent monopoles (52); they represent a fractionalisation of the molecular cyanide ion and must interact via a Coulomb potential (6) “

This seems insipid to me, or am I missing something ? Of course one could think of these as emergent monopoles, but it is much easier just to think of them as what they are, electric charge, and of course electric charges interact via a coulomb potential. If they did not, then that would be very interesting.

Reviewer #1 (Remarks to the Author):

In this manuscript a system of frustrated electric dipoles (“dimers”) in $\text{Cd}(\text{CN})_2$ -- argued to be analogous to the physics of magnetic dipoles in spin ice materials -- is presented. Due to the larger energy scale of the electric dipole interaction (vs. the magnetic dipole interaction) elements of this physics appear even at room temperature, much higher than the $O(1\text{K})$ scales typical for the rare-earth based spin ice systems. The presence of this “spin ice” physics is primarily supported by a combination of neutron PDF and Cd NMR measurements, which indicate that the population of local CN dimer configurations tracks well with expectations based on an spin ice (Ising) model down to $\sim 130\text{K}$ or so. The dimer-dimer interactions being “ice-like” is supported by quantum chemical calculations. Below $\sim 130\text{K}$ a structural transition into an anti-ferroelectric state prevents exploration of any low temperature aspects of this ice physics. Some elaboration on the spin ice analogy is made through comparison of single crystal X-ray diffraction to theoretical expectations, as well as modelling what neutron scattering would reveal in a magnetic system analogous to $\text{Cd}(\text{CN})_2$.

Overall my impression of this work is quite positive. This material $\text{Cd}(\text{CN})_2$ presents an important opportunity to explore frustration (and spin ice physics in particular) at much higher temperature regimes and outside the usual set of magnetic materials. It will thus likely be of interest to those studying spin ice physics and frustrated magnets more broadly, as well as those interested in these kinds of structurally disordered/correlated materials.

The key result seems to be the characterization of this material above its structural transition. The manuscript makes a fairly convincing case that the interactions between the dimers are ice-like and that these dimers are at least somewhat dynamic above $\sim 100\text{K}$ or so. The theoretical analysis is consistent with this, though it is not so clear how reliable these kinds of quantum chemical calculations are in estimating these parameters. The range of methods presented in the SI make it clear that different ab-initio methods agree qualitatively, but not quantitatively, especially for the dipolar energy scale.

Unfortunately, the (structural) phase transition at $\sim 130\text{K}$ cuts off access to the low temperature physics, with the paraelectric phase disappearing at temperature scales comparable to the effective Ising exchange in the spin ice model, due to the presence of the two pyrochlore sublattices which are coupled through (at least) dipolar interactions $\sim O(100\text{K})$. This is a major difference from spin ice materials, where the limiting factor is the intra-sublattice dipolar interaction which, on its own in $\text{Cd}(\text{CN})_2$, would only result in ordering at much lower temperatures $\sim O(10\text{K})$. In other words this model, with its two sublattices, is much less frustrated than the usual dipolar spin ice.

While this is a real deficiency, I do not think it is a fatal one, as there are many potential avenues in perturbing or modifying this material that might alleviate this issue (e.g., as discussed by the authors, applying pressure, elemental substitution, etc). In addition, this two sublattice model has not been studied in much detail, and so may hold some surprises.

The authors further raise a number of interesting questions regarding the dynamics of the CN dimers and the relation of the conventional phonon modes to the unconventional excitations of the “ice” degrees of freedom (in this case the electric monopoles). These are likely to provide good starting points for future experimental and theoretical studies of these types of systems. I think the comments about single-network $\text{Cd}(\text{CN})_2$ are particularly promising, since the issue with ordering might be avoided (or at least strongly suppressed). The relation of this physics to its observed NTE is also intriguing.

In terms of the novel results demonstrated in the work, its appeal to a broad range of researchers and its potential to motivate further work, I believe this satisfies the criteria for publication in Nature Communications and thus recommend acceptance.

Thank you!

Reviewer #2 (Remarks to the Author):

The authors consider the phase behaviour and order associated with CN- orientations as well as Cd displacements to argue that Cd(CN)₂ likely displays phenomena akin to spin-Ice physics, but at relatively high temperature compared with that displayed by well-studied magnetic insulators with spin-Ice ground states.

This is a very interesting analogy to consider and expand upon, with the orientation of the CN- molecules playing the role of the large, Ising-like magnetic dipole moments in magnetic insulators such as Ho₂Ti₂O₇ and Dy₂Ti₂O₇. This geometrically frustrated ground state has generated much excitement, both in the classical spin Ice problem and its quantum analogue, quantum spin ice. Of course, it is not so surprising that a structural analogue to such a magnetic problem could exist, as our understanding of these magnetic insulators is based on analogy with the physics of "real" water ice.

As the authors discuss, the smoking gun signatures for spin ice correlations and spin ice physics in magnetic insulators have been the observation of pinch-point diffuse scattering in single crystal scattering experiments and a residual Pauling entropy associated with the heat capacity in these systems. I think it is fair to say that were convincing data of this form available and presented, we would likely be discussing a contribution to Nature.

As it is, such smoking gun signatures are not available, at least in part because single crystals appropriate for neutron scattering studies are not available, and the temperature scale of the phenomenon in Cd(CN)₂ is as high as it is. Nonetheless, this is a very well written manuscript that sets out the case that indeed CN- flips do occur, and that some orientational order of the CN- molecules develop in the temperature range of interest. They then map the available data onto a minimal effective spin Hamiltonian and show that the corresponding state describing Cd(CN)₂ should be related to classical spin ice over a reasonable high temperature regime. While I have a couple of questions, I believe that this is an interesting account of a new and topical problem, and that it is appropriate to Nature Communications, after the relatively minor points have been addressed.

Many thanks!

My relatively minor points are:

1- The authors show x-ray diffuse scattering attributed to displacements of the Cd and dependent on the details of the local CN- orientations around the Cd. Presumably the size of the Cd displacements was determined (perhaps this is in the supplemental info - if so I would highlight this in the main manuscript), and this should be explicitly mentioned.

A good point: it's about a half an Ångström, which is of course very significant. We have a strong handle on this from our previous single-crystal X-ray diffraction study [Ref. 38] and also our QC calculations. We have added a note to this effect in the main text.

I also assume that there are no corresponding displacements associated with the magnetic spin ice problem. Would such displacements, and/or local distortions of the tetrahedra influence the predicted diffuse neutron scattering and pinch points as seen in Fig. 4 e)?

Thanks for raising this point. One does indeed expect these distortions to occur, and they are also mixed up with the low-energy NTE phonon modes (transverse displacements of the CN- ions away from the Cd...Cd vectors). All these aspects — in addition to the Cd displacements — are taken into account in our generation of Fig. S12, which illustrates a representative slice of the single-crystal neutron scattering one might hope eventually to measure experimentally. Fig. 4(e) plays a different role, in that it shows the effective (fictitious) magnetic scattering one might expect to see for a magnet governed by the Hamiltonian (2) at 300 K with the values of J , D , Δ of relevance to Cd(CN)₂, and then to compare with that from other relevant experiments and calculations. We realise this is not made appropriately clear and so we have modified the text in the manuscript accordingly.

2- There has been significant recent interest in insulating pyrochlore magnets based on transition metal magnetism, rather than rare earth magnetism, and much of this has been motivated by the same motivation presented here - to access spin ice like physics at higher than Kelvin and sub-Kelvin temperatures. The system NaCaX_2F_7 with $\text{X}=\text{Co}^{2+}$ and Ni^{2+} have been recently studied, with some evidence for pinch point correlations in $\text{NaCaNi}_2\text{F}_7$. Of course, these systems have a disordered "A" pyrochlore sub lattice, nonetheless, their relevant temperature scales are much elevated compared with, say, $\text{Dy}_2\text{Ti}_2\text{O}_7$, and it may be worth mentioning these. The relevant publications I know of are:

K.A. Ross et al, PRB, 93, 104433, 2016; Plumb et al, Nature Physics, 15, 54, 2019; Zhang et al, PRL, 122, 167203, 2019.

In retrospect this is an obvious omission from our manuscript, and we are grateful to the referee for flagging this. We have now included a brief, but important, nod to these materials in our introduction.

Reviewer #3 (Remarks to the Author):

Spin-ice physics in cadmium cyanide.

This article describes x-ray and neutrons diffraction, MAS NMR measurements and Monte Carlo calculations on the NTE material $\text{Cd}(\text{CN})_2$. The motivation according to the authors is to find ice-like phases at more accessible temperatures. The manuscript is well organised and written.

Thank you.

However i cannot recommend publication in Nature Comm.

The basic premise of the article is misleading, that spin-ice physics (such as that seen in $\text{Dy}_2\text{Ti}_2\text{O}_7$ and $\text{Ho}_2\text{Ti}_2\text{O}_7$) occurs in this system. The authors do show very convincingly that proton ordering ice-rules and charge-ice dynamics are observed. However the experimental evidence used to map this onto the spin-ice behaviour is not sufficient to support the claim of "spin-ice" physics.

Not to mention that there is simply no spin present in $\text{Cd}(\text{CN})_2$. Granted we often use classical spin models, but nevertheless magnetic spin dipole moment is not equivalent to an electric dipole. Spin is coupled to angular momentum, defined by spin operators and quantum numbers m_i , obeys selection rules, parity, etc.

Why force a round peg into a square hole ?

There is perhaps a philosophical difference here that the usual to-and-fro of peer-review is unlikely ever to resolve. We have strong sympathy for the referee's arguments, and fully accept (of course) that the electric dipoles of CN^- ions in $\text{Cd}(\text{CN})_2$ are meaningfully different from the magnetic dipoles of rare-earth ions in the conventional spin-ices.

Yet this is a field with its roots in drawing analogies, some of which are more exact than others. As the referee points out, many aspects of spin-ice behaviour have historically been understood in the context of classical models based on vector pseudospins. The very early papers of Bramwell and co-workers are clear in this regard (e.g. refs [2] and [44] in the text). Moreover there are many different flavours of spin-ice: Ising and Heisenberg, classical and quantum, and so on. All of this is of course good and makes for the scientific richness of the field.

Whatever one's viewpoint, the Hamiltonian (2) in our manuscript — which we show accurately captures the behaviour of $\text{Cd}(\text{CN})_2$ over the temperature range we probe (the referee does not question this) — is closely related to that used to describe many *bona fide* spin-ices (more on this below). This is the fundamental reason why we believe it sensible to draw the analogy to spin-ices as we do.

here are a few major complaints:

1) After an excellent introduction, the authors then try to describe $\text{Cd}(\text{CN})_2$ using the Heisenberg Hamiltonian of ref 44. But immediately this does not work, because $\text{Cd}(\text{CN})_2$ consist of two identical interpenetrating pyrochlore lattices, with each ion thus feeling twice as many nearest neighbours, and dipole forces along different directions.

This is only partly the case, and we apologise for any confusion here. The ‘exchange’ term operates only between neighbours of the same pyrochlore lattice, and so this does map directly onto the Hamiltonian of [44]. It is the dipolar term that links the two frameworks. We are up-front about this in the manuscript, and show that it is this additional dipolar contribution that unfrustrates the system and leads to the long-range order below 130 K.

2) In one of the most important papers (ref 14), the canonical spin-ice system $\text{Dy}_2\text{Ti}_2\text{O}_7$ was found to be an analog to cubic ice when heat capacity measurements showed an excess entropy that could be explained by the ice-rules first described by Pauling. The authors explain that this cannot be measured because there is a structural phase transition at 130K, and thus cannot enter deep into the “spin-ice” phase, presumably to integrate the heat capacity. This is a pity. But even if it were possible, this would only show that $\text{Cd}(\text{CN})_2$ obeys the ice-rules.

We agree with all the points made here by the referee. In due course it may be possible to make meaningful residual entropy measurements with single-network $\text{Cd}(\text{CN})_2$, but that isn't possible here for the reasons discussed already and made clear in the manuscript.

3) For the same reasons as above, and the fact that large single crystals are not available, magnetic diffuse scattering experiments could not be made in order to observe the “pinch-points” as seen $\text{Dy}_2\text{Ti}_2\text{O}_7$ and $\text{Ho}_2\text{Ti}_2\text{O}_7$. But even if there was a large single crystal, and for some reason there was no phase transition, what would magnetic diffuse scattering reveal on a sample that is simply not magnetic ? (not to mention the problems to measure it with Cd neutron absorption!)

Nevertheless the authors offer as “proof” Monte Carlo simulations of what magnetic diffuse scattering might look like compared with other systems in figure 4e. In particular they compare the simulation with “magnetic diffuse scattering” of $\text{Dy}_2\text{Ti}_2\text{O}_7$ and $\text{Pr}_2\text{Zr}_2\text{O}_7$. But the scattering pattern shown for $\text{Dy}_2\text{Ti}_2\text{O}_7$ is not magnetic diffuse scattering, it is just diffuse scattering, and for $\text{Pr}_2\text{Zr}_2\text{O}_7$ the measurements were at fixed energy 0.25meV, so it is not the same thing. Finally for Monte Carlo results they adopt the magnetic form factor of Dy. This makes no sense.

The referee has quite justifiably flagged in their point 1 above the question of whether the Hamiltonian at the heart of our study [Eq. (2)] drives behaviour at all related to that of the spin-ices. One way of assessing this is to compare the pairwise spin correlation functions that develop at finite temperature. Visualisation of the function $S_{\text{mag}}(\mathbf{Q})$ is an efficient way of capturing these correlation functions and their orientational dependence in frustrated magnets. This is of course one of the reasons why many in the spin-ice community attach particular importance to the experimental measurement and interpretation of this function, and why the ‘pinch point’ patterns are so iconic.

To make such a comparison as straightforwardly as possible we need to interpret the CN-pseudospins of our MC simulations as if they were magnetic dipoles, and to associate them with an appropriate magnetic form factor (we chose that of Dy^{3+} because we include data for $\text{Dy}_2\text{Ti}_2\text{O}_7$, but that of Pr^{3+} or of any lanthanide would give essentially identical results). In this way we can calculate an effective $S_{\text{mag}}(\mathbf{Q})$ — that although it can never be measured can nonetheless be compared with computed or measured $S_{\text{mag}}(\mathbf{Q})$ functions for conventional spin-ices.

An obvious test is against the Hamiltonian of Ref. 44, which leads to the $S_{\text{mag}}(\mathbf{Q})$ function illustrated in the bottom-left corner of Fig. 4(e). That this function is qualitatively similar to that derived from our $\text{Cd}(\text{CN})_2$ configurations for the same effective temperature ($T \sim 1.5 J_{\text{eff}}$) tells us that the two models are behaving in similar ways at this temperature — despite the additional inter-framework dipolar contribution discussed above. Quite understandably there are relatively

few experimental measurements of $S_{\text{mag}}(\mathbf{Q})$ in spin-ices reported in the literature this same effective temperature, since it is out of the deep spin-ice regime. The data we include for $\text{Dy}_2\text{Ti}_2\text{O}_7$ (top right panel) are representative of the famous pinch-point modulations one sees within this regime. (NB we use “magnetic diffuse scattering” because this diffuse scattering arises from the magnetic, rather than nuclear, scattering contribution). At higher temperatures one populates the spin excitations ordinarily probed with inelastic scattering, which is why we include the inelastic measurements for $\text{Pr}_2\text{Zr}_2\text{O}_7$. This is relevant because it links the fluctuations in representative spin-ices to the finite- T behaviour of our $\text{Cd}(\text{CN})_2$ model.

We hope this explains our rationale here, and we have included some additional clarification in the accompanying text.

We worry the referee has conflated a few other ideas within this point as well, that we now address:

For clarity, we make no claim that $\text{Cd}(\text{CN})_2$ would give magnetic scattering in a neutron scattering measurement. It has no unpaired electrons. We do calculate the expected (nuclear) single-crystal neutron scattering pattern we would expect to observe for $\text{Cd}(\text{CN})_2$; it is shown in Fig. S12.

We are also keenly aware of the problems of neutron absorption by natural-abundance Cd. That is why we have gone to such extreme lengths to make a sample of enriched $^{114}\text{Cd}(\text{CN})_2$ suitable for neutron scattering measurements. The data shown in Figures 2 and 4(c) come from these measurements. We explain in the text why it is not possible to make these same measurements reliably using X-ray scattering, so this is an absolutely key aspect of our study. We do hope in due course to be able to carry out single-crystal neutron scattering measurements.

4) In this paper, the dynamics have been probed by exchange NMR spectroscopy (EXSY) described in some detail in the SI. From the analysis, thermal activation of the CN dipoles is deduced. However the results are based on only three temperatures 62, 67 and 85C ! Why ? In the main text it is mentioned “quite severe experimental constraints (see SI)” but this is not explained. Is the relaxation too difficult to measure? is it long below 60 C? too fast above 80 C ?

We apologise for not having expanded sufficiently on this point in the SI and have now added some additional text. The key factors are the following. These EXSY measurements are extremely time-consuming, especially as temperature is reduced. The lower bound of 60 °C arises from the long spin–lattice (T_1) relaxation time of ^{113}Cd in $\text{Cd}(\text{CN})_2$, which is such that obtaining suitable statistics in the build-up curves for lower temperatures was clearly impractical (further individual measurements would take months of continuous instrument time). The probe itself is not designed to operate at temperatures above 85 °C. In addition, this temperature is also close to the limit of the measurement technique for this sample, since further increase in jump rates will lead to flips during the initial selective excitation and so prevent a meaningful result. Taken together this is why we have had to content ourselves with relatively few measurements within a narrow temperature window.

This is the main experimental conclusion of the paper, that thermal activation in $\text{Cd}(\text{CN})_2$ is similar to spin-ice.

But this is by no means conclusive.

These various constraints described above translate to an increased experimental uncertainty regarding the flipping barrier height Δ (effectively the slope of the line in Fig. 3(d)). We feel the text as it stands is already sufficiently cautious regarding our interpretation of the numerical value obtained and we know also that sensible variations in this value do not affect the fundamental physics we report.

What is conclusive is the thermal activation of CN-flips, which is directly evident in the observation of off-diagonal contributions to the EXSY spectrum. This is the point made in Fig. 2(e).

The dynamics in spin-ice is fascinating and has been instrumental in understanding the underlying physics. The dynamics of canonical spin-ice materials have been probed by ac-susceptibility and

relaxation measurements over many decades in frequency and from room temperature down milli kelvin temperatures. There is indeed thermal activation at high temperature, then as temperature is lowered a plateau develops where tunnelling is presumably occurring, then at lowest temperatures the dynamics are dominated by emergent magnetic monopoles. There is no analog with $\text{Cd}(\text{CN})_2$

Whether there exists a strict mapping to spin-ices or not, we anticipate that $\text{Cd}(\text{CN})_2$ will show a similarly rich spectrum of dynamical behaviour, characterisation of which we hope our present study will motivate in the future. In support of this statement we make the following observations. First, we know already that there is strong interplay between CN-flips and the conventional lattice dynamics since the 130 K displacive transition is driven by the former. Second, there is a long history of dielectric spectroscopy as an efficient probe of anomalous dielectric relaxation and excitation processes in the closely related families of (e.g.) KCN-based quadrupolar glasses. Third, tunnelling can play a role in CN-reorientations; James Sethna wrote a very nice survey of this phenomenon in a piece for the New York Academy of Sciences [484, 130 (1986)]. And, fourth, there is no *a priori* reason why emergent monopoles in $\text{Cd}(\text{CN})_2$ might not also play a role in its dynamics.

5) Finally the authors mention that “the C-rich or N-rich Cd coordination environments assume the role of emergent monopoles (52); they represent a fractionalisation of the molecular cyanide ion and must interact via a Coulomb potential (6) “

This seems insipid to me, or am I missing something ? Of course one could think of these as emergent monopoles, but it is much easier just to think of them as what they are, electric charge, and of course electric charges interact via a coulomb potential. If they did not, then that would be very interesting.

We agree, and have toned down the text accordingly.

REVIEWERS' COMMENTS

Reviewer #1 (Remarks to the Author):

My recommendation that the manuscript should be published in Nature Communications remains unchanged.

The two other reviewers have raised several points and questions (I did not have any serious concerns of my own). The concerns of reviewer #2 were mostly minor, and I think it is clearly the authors answered satisfactorily.

The more significant issues raised by reviewer #3 were of a philosophical nature, specifically with regard to merit and value of finding "analogies" between physical systems. While I am sympathetic to some of the views they express, I would side with the authors in their argument that the physics of Cd(CN)₂ is sufficiently similar, but also sufficiently distinct, to merit (a) further study and (b) publication in Nat. Comm. Indeed, given spin-ice itself was named and motivated in analogy to water ice, I do not think this is disqualifying.

With respect to the more technical concerns raised by reviewer #3, I think the responses of the authors are sufficient in light of some of the experimental challenges.

Reviewer #2 (Remarks to the Author):

My original review of this manuscript was quite positive with three relatively minor criticisms brought forward. The authors have satisfactorily addressed the three points that I raised. In addition, I have read through the criticisms and associated replies by the authors to the two other reviewers of this manuscript, and again my feeling is that the authors have done their due diligence in addressing these points as well. For that reason I now recommend acceptance of this manuscript, in its present form, for publication in Nature Communications.